# Towards Scalable Semantic Representation for Recommendation

## Abstract

With recent advances in large language models (LLMs), there has been emerging numbers of research in developing Semantic IDs based on LLMs to enhance the performance of recommendation systems. However, the dimension of these embeddings needs to match that of the ID embedding in recommendation, which is usually much smaller than the original length. Such dimension compression results in inevitable losses in discriminability and dimension robustness of the LLM embeddings, which motivates us to scale up the semantic representation. In this paper, we propose Mixture-of-Codes, which first constructs multiple independent codebooks for LLM representation in the *indexing stage*, and then utilizes the Semantic Representation along with a fusion module for the downstream *recommendation stage*. Extensive analysis and experiments demonstrate that our method achieves superior discriminability and dimension robustness scalability, leading to the best scale-up performance in recommendations.

## 1 Introduction

Recently, the emergence of large language models (LLM) (Dubey et al., 2024; Achiam et al., 2023) sheds light in improving the recommendation systems via the semantic knowledge from LLM (Hou et al., 2024; Bao et al., 2023). An intuitive practice is to simply project the LLM embeddings to low-dimension embeddings via only MLPs into the recommendation systems for feature interactions. However, such application is ineffective, largely due to the massive semantic gap between the embedding spaces of LLM and recommendation systems (Lin et al., 2023; Pan et al., 2024).

Several works (Rajput et al., 2024; Singh et al., 2023) have proposed to derive Semantic IDs (*i.e.*, codes) based on clustering methods such as VQ-VAE (Van Den Oord et al., 2017) or RQ-VAE (Lee et al., 2022) to capture information from the LLM embedding. In particular, they first train an auto-encoder with discrete codes and then apply these codes to downstream tasks such as retrieval or ranking. Such methods aim to transfer knowledge from the LLM embedding space to recommendation systems, utilizing the codes to capture the local structure of the original space. Besides, embedding these codes in the downstream stage facilitates the effective training in an end-to-end manner.

Notably, the LLM embeddings usually have very large dimensions, ranging from 4,096 to 16,384 (Dubey et al., 2024). When generating the embeddings for these codes, their dimension needs to match that of the recommendation IDs. However, the dimension in recommendation is usually small due to the *Interaction Collapse Theory* (Guo et al., 2023). Therefore, the code embeddings are also only able to span a low-dimension space. With one single semantic embedding as the semantic representation, it may fail to capture the complex, high-dimensional structure of the original LLM embeddings, *lead to inevitable information loss and performance deterioration* during the knowledge transfer. This motivates us to study how to *scale up the semantic representation effectively*.

We delve into two approaches based on existing works, including Multi-Embedding (Guo et al., 2023) and RQ-VAE (Lee et al., 2022), to scale up the semantic representation by either using one codebook with multiple embeddings, or multiple hierarchical codebooks and embeddings. Nevertheless, the analysis and empirical results demonstrate that the representations of these two methods are not scalable in terms of discriminability and dimension robustness.

In this paper, we propose Mixture-of-Codes (MoC), a novel two-stage approach to effectively scale up semantic representations for recommendation. First, we propose a Multi-Codebooks VQ-VAE

method that learns multiple independent discrete codebooks in the indexing stage. Once we have these codes for all items, we adopt a Mixture-of-Codes module to fuse the learnable embeddings of multiple codes in the downstream recommendation stage. We use the name MoC to refer both the second stage itself, as well as the whole two-stage approach. Comprehensive analysis shows that our method successfully achieves scalability regarding discriminability, dimension robustness, and performance. Our contributions can be summarized as follows:

- We pioneer a study on the scalability of semantic representation on transferring knowledge from LLM to recommendation systems, and reveal that several baseline approaches fail to scale up effectively.
- We propose a novel two-stage Mixture-of-Codes approach, which learns multiple codebooks in the indexing stage based on LLM embeddings and then employs a Mixture-of-Codes module to fuse the embeddings of multiple codes in the downstream recommendation stage.
- Comprehensive experiments on three public datasets show that our method successfully achieves scalability regarding both discriminability, dimension robustness, and performance.

## 2 PRELIMINARIES

**VQ-VAE.** VQ-VAE first encodes the input $x$ with an encoder $\mathcal{E}$ and then train a codebook to transform embedding into discrete tokens. Formally, a codebook $\mathcal{Z} = \{z_k\}_{k=1}^{K}$ is define as a finite set with prototype vectors $z_k \in \mathbb{R}^{n_z}$, where $K$ is the codebook size and $n_z$ is the dimensionality of code embeddings. Given the encoder output $\mathbf{z} := \mathcal{E}(x) \in \mathbb{R}^{n_z}$, VQ-VAE quantities the embedding with the code whose embedding is nearest to $\mathbf{z}$, that is,

$$\mathbf{z}^{\mathbf{q}} = \arg\min_{z_k \in Z} \|\mathbf{z} - z_k\|_2^2. \tag{1}$$

Then the reconstruction is derived based on the quantized output $\mathbf{z_q}$ and a decoder $\mathcal{D}$: $\hat{x} = \mathcal{D}(\mathbf{z_q})$. The model and codebook can be trained end-to-end via the loss function

$$\mathcal{L}_{\text{VQ}}(\mathcal{E}, \mathcal{D}, \mathcal{Z}) = \|x - \hat{x}\|^2 + \|\text{sg}[\mathbf{z}^{\mathbf{q}}] - \mathbf{z}\|_2^2 + \|\text{sg}[\mathbf{z}] - \mathbf{z}^{\mathbf{q}}\|_2^2, \tag{2}$$

where sg[·] denotes the stop-gradient operation, the first term $\mathcal{L}_{\text{rec}} = \|x - \hat{x}\|^2$ is a reconstruction loss, the second term is the commitment loss that is used to force the encoder output $\mathbf{z_q}$ commits to the codewords and the bottleneck codewords are optimized by the third term. In practice, we perform moving averages update (Van Den Oord et al., 2017) instead of adding auxiliary losses for stable training of the codebook. Then the selected index can be used as Semantic IDs for clustering in the context of the semantic codebook, therefore capturing local structure of the original embedding.

**Semantic IDs for Feature Interaction.** In recommendation system, feature interaction models how different attributes from users and items influence each other to affect recommendation outcomes. When incorporating the Semantic ID from quantization, the models treat the Semantic ID $x_{sid}$ as a new feature field. The features are fed with other $N$ features into the embedding layer $E_i$ for each field and subsequently into the feature interaction modules for prediction.

$$\begin{aligned} \boldsymbol{e}_i &= \boldsymbol{E}_i^{\top} \mathbf{1}_{x_i}, \; \forall i \in \{1, 2, ..., N\}, \\ \boldsymbol{e}_{sid} &= \boldsymbol{E}_{sid}^{\top} \mathbf{1}_{x_{sid}}, \\ h &= I(\boldsymbol{e}_1, \boldsymbol{e}_2, ..., \boldsymbol{e}_n, \boldsymbol{e}_{sid}), \\ \hat{y} &= F(h), \end{aligned} \tag{3}$$

## 3 ON THE SCALING OF SEMANTIC REPRESENTATION

In this section, we first revisit the design of one single codebook for recommendation and discover the information loss due to dimension compression. Based on this observation, we are motivated to design scalable semantic representations and propose two quantitative metrics to measure the information and dimension gain from scaling. Next, we conduct a detailed analysis of existing scaling methods based on these metrics.

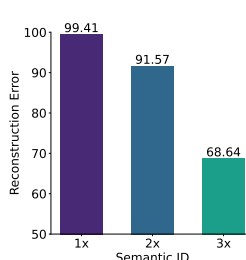

Figure 1: Reconstruction Error with different sets of Semantic IDs.

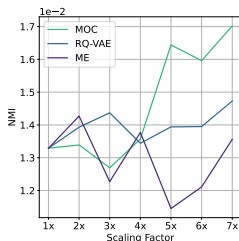

(a) Flatten Emb NMI w.r.t. scaling factors, with 100 clusters.

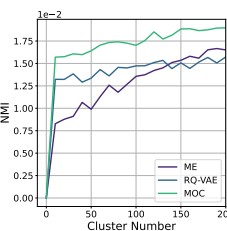

(b) Flatten Emb NMI w.r.t. Cluster Number, with scaling factor of 7x.

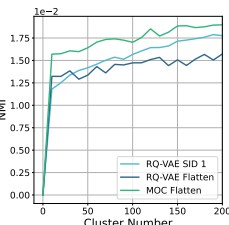

(c) Comparison with RQ-VAE SID 1 embedding, with scaling factor of 7x.

Figure 2: Scalability on discriminability of various methods.

### 3.1 LIMINATION OF A SINGLE CODE

The original LLM embeddings span a high-dimensional space, *e.g.*, ranging from 1,024 to 416,384 dimensions. (Dubey et al., 2024). When we build a semantic representation based on the LLM embeddings to transfer their knowledge to recommendation, the dimension of these representation needs to match that of the recommendation ID embeddings. However, the dimension of recommendations is usually no more than 256 due to the Interaction Collapse Theory (Guo et al., 2023), and hence is much smaller than that of the LLM embeddings. Such dramatic dimension compression may result in huge information loss.

To illustrate this empirically, we conduct a toy study based on the reconstruction error of the LLM embeddings in a VAE-based generative model. As shown in Fig 1, we utilize a two-layer MLP with 512 hidden units to reconstruct the original 4096-dimensional LLM embedding. When using only single set of Semantic ID as input, the reconstruction error is extremely high, indicating the significant information loss. However, the error drops significantly when we scale up the dimensions via using multiple (*i.e.*, 2x and 3x) independent codebooks. This demonstrates the original single code embedding only preserves limited information of the LLM embeddings. Therefore, we aim to scale up the semantic representation appropriately to preserve the rich information from the LLM, thereby improving the performance of downstream recommendation tasks.

### 3.2 BASELINE APPROACHES TO SCALABLE SEMANTIC REPRESENTATION

Below we present two baseline approaches to scale up semantic representation based on Multi-Embedding (Guo et al., 2023) and RQ-VAE (Lee et al., 2022).

**Single Codebook with Multi-Embeddings.** Inspired by the recent Multi-Embedding (Guo et al., 2023) approach to scale up embeddings in recommendation systems, our first choice is to assign multiple embeddings for each semantic ID. Specifically, we still learn only one single codebook during the indexing stage, while we build $M$ independent embeddings $\{e_{sid}^1, \ldots, e_{sid}^M\}$ in the downstream stage. Formally, we have

$$
\begin{aligned}
e_{sid}^i &= (\boldsymbol{E}_{sid}^i)^\top \mathbf{1}_{x_{sid}}, \ \forall i \in \{1, 2, ..., M\}, \\
h &= I(\boldsymbol{e}_1, \boldsymbol{e}_2, ..., \boldsymbol{e}_n, \boldsymbol{e}_{sid}^1, ..., \boldsymbol{e}_{sid}^M).
\end{aligned}
\tag{4}
$$

**RQ-VAE (*i.e.*, Multiple Hierarchical Codebooks and Multi-Embeddings).** RQ-VAE is a common practice for deriving Semantic IDs in recommendation systems (Rajput et al., 2024; Jin et al., 2023; Zheng et al., 2024). It applies quantization on residuals at multiple levels with different codebooks. The reconstructed target in the next level is the residual representation in the current level:

$$
\begin{aligned}
\mathbf{z_i}^{\mathbf{q}} &= \arg\min_{z_k \in Z_i} \|\mathbf{z_i} - z_k\|_2^2, \\
\mathbf{z_{i+1}} &= \mathbf{z_i} - \mathbf{z_i}^{\mathbf{q}}.
\end{aligned}
\tag{5}
$$

Due to the hierarchical design of RQ-VAE, the Semantic IDs obtained from the codebooks are highly dependent and entangled. With $M$ levels of hierarchical Semantic IDs $\{x_{sid_i}\}_{i=1}^M$, it is practical to

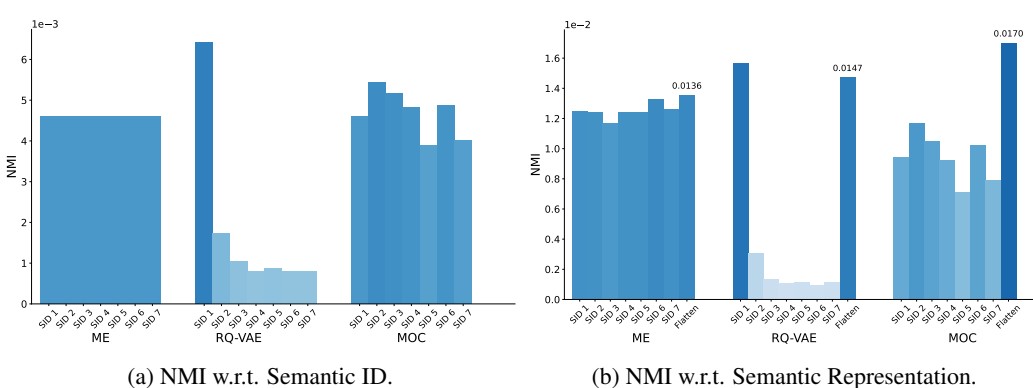

(a) NMI w.r.t. Semantic ID.

(b) NMI w.r.t. Semantic Representation.

Figure 3: Normalized Mutual Information(NMI) of Semantic Representation with 7x scaling factor.

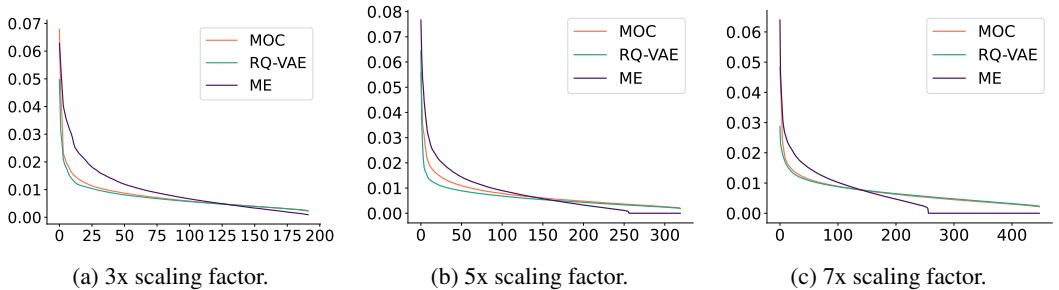

(a) 3x scaling factor.

(b) 5x scaling factor.

(c) 7x scaling factor.

Figure 4: Scalability of Dimension Robustness regarding different scaling factors. Each figure presents the singular spectrum of the semantic representation at the given scaling factor.

utilize to scale the Semantic Representation as follows.

$$\boldsymbol{e}_{sid_i} = (\boldsymbol{E}_{sid_i})^\top \mathbf{1}_{x_{sid_i}}, \ \forall i \in \{1, 2, ..., M\}$$

$$h = I(\boldsymbol{e}_1, \boldsymbol{e}_2, ..., \boldsymbol{e}_n, \boldsymbol{e}_{sid_1}, ..., \boldsymbol{e}_{sid_M}). \tag{6}$$

### 3.3 MEASUREMENT ON SCALABILITY OF SEMANTIC REPRESENTATION

Below we present two ways to quantify the scalability of semantic representations, one from a discriminability perspective and another from a dimension robustness perspective.

**Definition 3.1** (Discriminability Scalability of Semantic Representation). With each scaling factors from 1x to $Mx$, the discriminability of a Semantic Representation in the continuous space is defined as the mutual information between its quantized representation $Q(\boldsymbol{r})$ and the supervised label in the downstream tasks $Y$, *i.e.*, $\mathrm{MI}(Q(\boldsymbol{r}), Y)$.

Following previous approach (Jawahar et al., 2019), we apply K-means as the discrete method for normalized mutual information (NMI) calculation and analyze the flatten embedding of the concatenated Semantic Representations among different methods in Fig. 2. We present results with respect to different scaling factors with 100 clusters and provide flattened embedding NMI under varying cluster numbers in Fig. 1b. We observe that the discriminability of the ME does not increase with the scaling factor and may even decrease slightly. This is because all these extra embeddings still correspond to the same Semantic IDs from a single codebook, thus containing minimal additional information and redundancy.

Regarding RQ-VAE, its discriminability also does not consistently increase with the scaling factor due to the fact that the additional fine-grained Semantic IDs introduced at higher level contain diminishing information. We illustrate this by comparing NMI across different Semantic IDs and their representations in downstream tasks in Fig. 3. Another surprising observation from the results in Fig. 3 is that the lowest level Semantic ID, *i.e.*, SID 1, and its representation contain more information

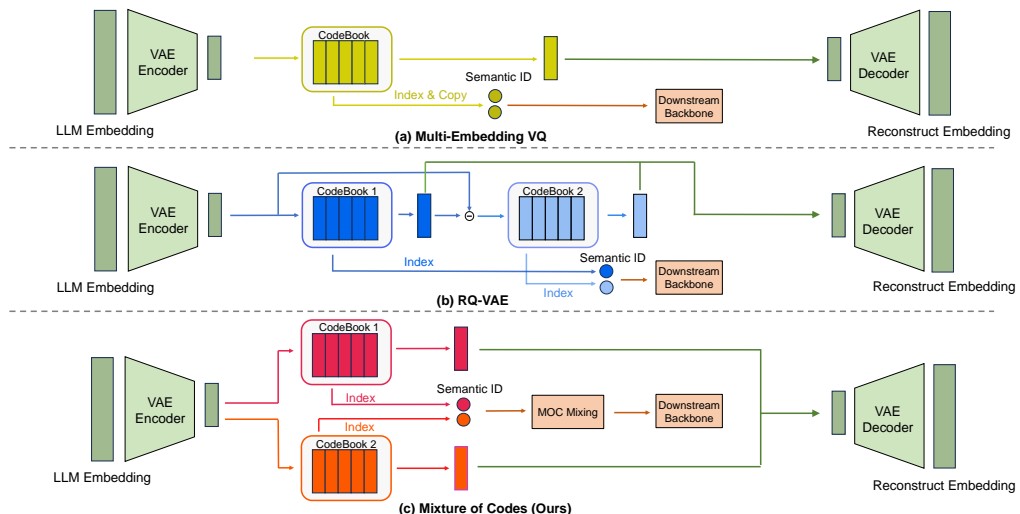

Figure 5: **Comparison among Multi-Embedding VQ, RQ-VAE and Mixture-of-Codes. The codebooks with deeper color contain more information relevant to the input data. (a) Multi-Embedding VQ builds independent embeddings for a single set of semantic IDs and is equivalent to perform index copying for downstream models. (b) RQ-VAE utilizes hierarchical codebooks and high-level semantic IDs are less informative. (c) Our Mixture-of-Codes uses parallel codebooks to capture important semantics in the original LLM space and employs a fusion network for better generalization in downstream tasks.**

than the flattened embedding of all the Semantic IDs. We provide the NMI results across various cluster numbers in Fig. 1c and observe that the NMI of SID 1 is consistently larger than that of the flattened embedding when the cluster number exceeds 50, demonstrating that higher-level Semantic IDs may hinder the generalization of lower-level Semantic IDs.

When scaling up the Semantic Representation, the interaction between low-frequency information and high frequency becomes important as the dimension increases. Therefore, we propose a new metric to measure the dimension robustness of the scaling Semantic Representation.

**Definition 3.2** (Dimension Robustness Scalability of Semantic Representation)**.** The dimension robustness scalability of Semantic Representation can be measured by the singular spectrum of the Semantic Representation under different scaling factors. A robust Semantic Representation should have higher top singular values without suffering from dimension collapse.

We plot the singular spectrum of ME and RQ-VAE in different scaling factors in Fig. 4 to compare the dimension robustness and its scalability between models. And we have the following observations.

**Observation 1.** RQ-VAE doesn't suffer from dimensional collapse since that its long-tail singular values don't diminish. However, its top singular values are not large enough compared with ME.

**Observation 2.** ME has the largest top singular values. However, it suffers from dimensional collapse since its long-tail singular values diminishes suddenly after index 250 in 5x and 275 in 7x setting.

We conclude with the following finding:

> *Finding 1. Existing methods such as ME and RQ-VAE are not scalable semantic representations for recommendation regarding discriminability and dimension robustness.*

## 4 METHOD

In this section, we propose Mixture-of-Codes (MoC) as a novel two-stage method to scale up semantic representation. We first introduce multiple codebooks in the indexing stage to generate multiple sets of Semantic IDs, and then present Mixture-of-Codes in the downstream recommendation modeling stage for better knowledge transfer.

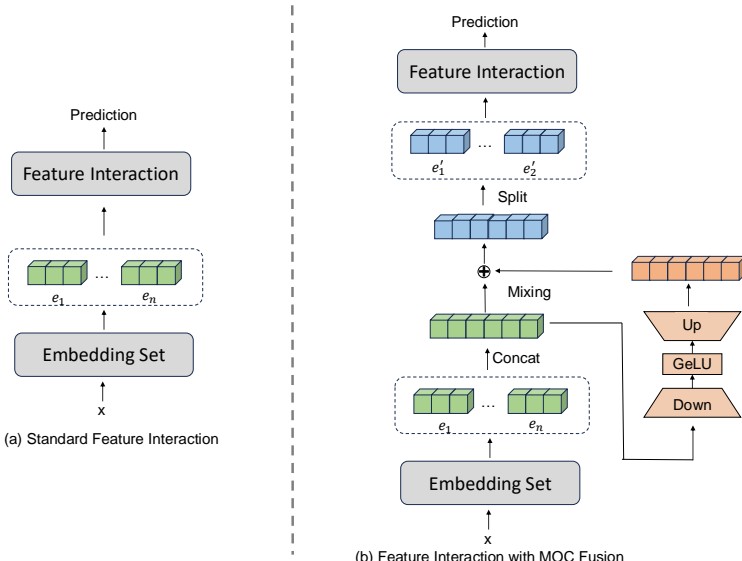

Figure 6: **The overall architecture of MoC Fusion. A bottleneck network is adopted for feature fusion in the downstream stage.**

### 4.1 MULTI CODEBOOKS FOR VECTOR QUANTIZATION

Motivated by the observation that multi-embedding does not provide new information due to the same Semantic ID while RQ-VAE focuses on hierarchical indices that are less informative for downstream tasks, we aim to uncover more information from the LLM embedding at a more fundamental level. Specifically, instead of the hierarchical design of RQ-VAE, we utilize multiple parallel codebooks to capture complementary information of the LLM embedding. Different codebooks project the hidden embedding into various spaces and collaborate to extract information from the LLM embedding. Formally, given encoder $\mathcal{E}$, decoder $\mathcal{D}$ and $N$ Codebooks $\{\mathcal{Z}_i\}_{i=1}^N$, we perform an average over their quantized embedding and the training loss is

$$\mathcal{L}_{\text{MoC}}(\mathcal{E}, \mathcal{D}, \{\mathcal{Z}_i\}_{i=1}^N) = \|x - \hat{x}\|^2 + \|\text{sg}[\mathbf{z^q}] - \mathbf{z}\|_2^2 + \|\text{sg}[\mathbf{z}] - \mathbf{z^q}\|_2^2,$$
$$\mathbf{z^q} = \text{AVG}(\{\mathbf{z_i^q}\}_{i=1}^N),$$

(7)

where $\mathbf{z}^q$ is the average quantization results and we select the corresponding indices as Semantic IDs.

### 4.2 MIXTURE-OF-CODES FOR IMPLICIT FUSION

For traditional mixtures of experts, a gating router is used to select some of the experts and perform mixing based on the weights generated by the router with the help of the task-specific loss. However, this approach is impractical in MoC since we do not train the codebooks in an end-to-end style, and the embeddings are initialized and tuned in the downstream stage.

Therefore, we propose a fusion network in the downstream stage for implicit fusion of the codebooks. Specifically, we employ a bottleneck network following the embedding layer to ensure information flow across different features before the feature interaction modules, as shown in Figure 6. This implicit fusion design is trained using task-specific loss and mixes the embeddings for better performance, serving a role similar to the gating network. Formally, given $N$ original attributes and $M$ Semantic IDs, we have

$$e_{\text{concat}} = \text{CONCAT}(\boldsymbol{e}_1, ..., \boldsymbol{e}_n, \boldsymbol{e}_{\text{sid}_1}, ..., \boldsymbol{e}_{\text{sid}_M}),$$
$$e'_{\text{concat}} = e_{\text{concat}} + e_{\text{concat}} \cdot \mathbf{W}_{\text{down}} \cdot \mathbf{W}_{\text{up}},$$
$$\boldsymbol{e}_1, ..., \boldsymbol{e}_n, \boldsymbol{e}_{\text{sid}_1}, ..., \boldsymbol{e}_{\text{sid}_M} = \text{SPLIT}(e'_{\text{concat}}),$$

(8)

where $\mathbf{W}_{\text{down}}$ and $\mathbf{W}_{\text{up}}$ denotes the down and up projection layer, respectively.

Table 1: Test AUC of different methods over various models. We report the test AUC results with 1x, 2x, 3x and 7x scaling factors.

| Model | | Toys | | | | Beauty | | | | Sports | | | |
|---|---|---|---|---|---|---|---|---|---|---|---|---|---|
| | | 1x | 2x | 3x | 7x | 1x | 2x | 3x | 7x | 1x | 2x | 3x | 7x |
| DeepFM | ME | | 0.7403 | 0.7397 | 0.7390 | | 0.6651 | 0.6649 | 0.6638 | | 0.6942 | 0.6928 | 0.6917 |
| | RQ-VAE | 0.7406 | **0.7409** | 0.7405 | 0.7398 | 0.6651 | **0.6676** | 0.6670 | **0.6687** | 0.6931 | **0.6945** | 0.6932 | 0.6937 |
| | MoC | | **0.7408** | **0.7415** | **0.7418** | | 0.6656 | **0.6674** | 0.6681 | | 0.6931 | **0.6936** | **0.6953** |
| DeepIM | ME | | 0.7396 | 0.7404 | 0.7395 | | 0.6620 | 0.6635 | 0.6637 | | 0.6907 | 0.6910 | 0.6925 |
| | RQ-VAE | 0.7404 | **0.7401** | 0.7403 | 0.7404 | 0.6648 | **0.6651** | 0.6660 | 0.6678 | 0.6931 | 0.6918 | 0.6925 | 0.6938 |
| | MoC | | **0.7401** | **0.7417** | **0.7422** | | 0.6641 | **0.6668** | **0.6691** | | **0.6927** | **0.6935** | **0.6942** |
| AutoInt+ | ME | | **0.7430** | 0.7419 | 0.7414 | | 0.6648 | 0.6630 | 0.6641 | | 0.6935 | 0.6930 | **0.6929** |
| | RQ-VAE | 0.7415 | **0.7430** | 0.7419 | 0.7418 | 0.6630 | **0.6672** | 0.6642 | 0.6677 | 0.6911 | 0.6934 | **0.6933** | 0.6915 |
| | MoC | | 0.7414 | **0.7420** | **0.7447** | | 0.6661 | **0.6651** | **0.6689** | | **0.6939** | 0.6926 | **0.6927** |
| DCNv2 | ME | | 0.7445 | 0.7449 | 0.7459 | | 0.6717 | 0.6716 | 0.6722 | | 0.6955 | 0.6963 | 0.6976 |
| | RQ-VAE | 0.7445 | 0.7457 | 0.7457 | 0.7469 | 0.6701 | **0.6719** | 0.6720 | 0.6726 | 0.6962 | 0.6965 | 0.6966 | 0.6979 |
| | MoC | | **0.7462** | **0.7458** | **0.7474** | | 0.6714 | **0.6730** | **0.6729** | | **0.6970** | **0.6972** | **0.6989** |

## 4.3 In-depth Scalability Analysis of MoC

**Scalability of Discriminability** We study the discriminability of MoC by the mutual information between quantized representation and the label at various scaling factors in Fig. 3b. It can be observed that the discriminability of MoC at 1x is comparable with that of RQ-VAE and much higher than ME. Furthermore, the discriminability of MoC gets higher with larger scaling factors from 1x to 7x, indicating that it has better scalability regarding discriminability.

**Scalability of Dimension Robustness** We plot the singular spectrum of MoC on various scaling factors in Fig. 4., and find that it gets higher values on the low-index singular, compared to the RQ-VAE, even though not as high as ME. Besides, its singular values on high-indices are also robust, not diminishing as ME in 5x and 7x factors. In conclusion, the dimension of MoC are more robust than ME and RQ-VAE when we scale up its representation.

Based on the observations above, we conclude with the following finding:

> *Finding 2. Our proposed MoC successfully enables scalable Semantic Representation regarding both discriminability and dimension robustness.*

## 5 Experiments

### 5.1 Setup

**Datasets.** We conduct experiments on three domains from Amazon review benchmark (He & McAuley, 2016): Amazon-Beauty, Amazon-Sports, and Amazon-Toys. We follow LMINDEXER (Jin et al., 2023) and keep the users and items that have at least 5 interactions to filter out unpopular interacting behavior. Given textual description of items comprising of title, brand and categories, we utilize LLM2Vec (BehnamGhader et al., 2024) with LLama3 (Dubey et al., 2024) as backbone to obtain their LLM embeddings. Early stop strategy are adopted over 8/1/1 training/validation/test splits of all the three datasets.

**Implementation Details.** We follow TIGER (Rajput et al., 2024) to set 256 as the codebook size and 32 as the latent representation. The encoder in the indexing stage has three hidden layers of size 512, 256 and 128 with ReLU activation. We evaluate the performance of DeepFM (Guo et al., 2017), DeepIM (Yu et al., 2020), AutoInt+ (Song et al., 2019) and DCNv2 (Wang et al., 2021) in the downstream tasks. We adpot the Adam optimizer with batch size 8012 and learning rate 0.001. All the experiments are run across three trials with different seeds and the averaged results are reported.

### 5.2 Overall Performance

We compare the three semantic scaling methods upon four representative CTR models, *e.g.*, DeepFM, DeepIM, AutoInt+ and DCN V2 on three public datasets. With the same number of IDs, our MoC

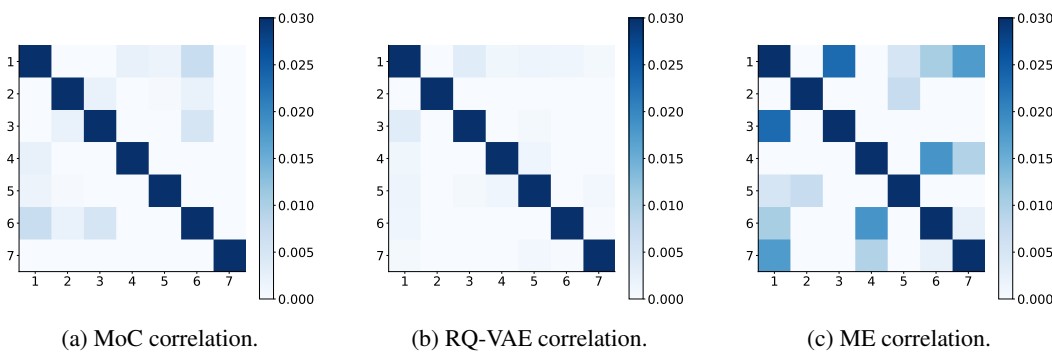

(a) MoC correlation.    (b) RQ-VAE correlation.    (c) ME correlation.

Figure 7: Correlation analysis of different methods.

outperforms the baselines by a large margin, especially in scenarios with a larger number of IDs, *e.g.*, 7x. Specifically, with 7x scaling factor, all the four models benefits from MoC on Toys datasets and surpass RQ-VAE by 0.20%, 0.18%, 0.29% and 0.05%, respectively.

More interestingly, in many scenarios, *our MoC succeed in achieving scaling law regarding the scaling factors*, *i.e.*, the performance increases when we include more Semantic Representations. In contrast, ME suffers from performance degradation due to the redundant semantic information, while RQ-VAE gain slight performance gain because of the less informative Semantic IDs at the high level.

### 5.3 CORRELATION BETWEEN MULTIPLE REPRESENTATION.

The Semantic Representation from different IDs have a deep influence on each other in downstream tasks, and hence we analyze the correlation between different representations to measure the extent of their similarity. Here we calculate Person correlation coefficient (Cohen et al., 2009) between Semantic Representation $e_{sid_i}$ and $e_{sid_j}$ over $n$ samples, and adopt dot product for multiplication of embedding vectors:

$$r_{ij} = \frac{\sum_{k=1}^{n}(e_{sid_i}^k - \bar{e}_{sid_i})(e_{sid_j}^k - \bar{e}_{sid_j})}{\sqrt{\sum_{k=1}^{n}(e_{sid_i}^k - \bar{e}_{sid_i})^2}\sqrt{\sum_{k=1}^{n}(e_{sid_j}^k - \bar{e}_{sid_j})^2}}.$$

As shown in Fig. 7, the Semantic Representation in ME are highly correlated with each other, *i.e.*, many different Semantic Representation have strong correlation, *i.e.*, representation 1 and 3, 4 and 6 are strongly correlated with each other. Such strong correlation makes the representation easily influenced by each other, leading to unstable optimization and ineffective scalability. Regarding MoC and RQ-VAE, the correlation between different Semantic Representation are low, as evidenced by the low correlation score in the off-diagonal cells in Fig. 7.

### 5.4 MORE COMPARISON RESULTS WITH RQ-VAE

To further verify the information contained in the semantic IDs at each level, we provide a more detailed comparison with RQ-VAE in terms of adding a single ID at each level in Fig. 8a and adding multiple IDs starting from the lowest level in Fig. 8b. As the results in Fig. 8a indicate, adding a single semantic ID at the low level of RQ-VAE provides a larger performance gain than at the high level, proving that semantic IDs at the high level hold little information. In contrast, MoC performs uniformly across various Semantic IDs and consistently show better performance than RQ-VAE. When equipping multi Semantic IDs starting from the lowest level, MoC gains significant improvements than RQ-VAE under different scaling factors, showcasing better generalization.

### 5.5 ABLATION ON MOC FUSION

We conduct an ablation study on MoC Fusion to verify the importance of mixing in the downstream stages. We equip both RQ-VAE and MoC with the fusion module and surprisingly find that both methods benefit from it when scaling up. We also examine the discriminability and dimension

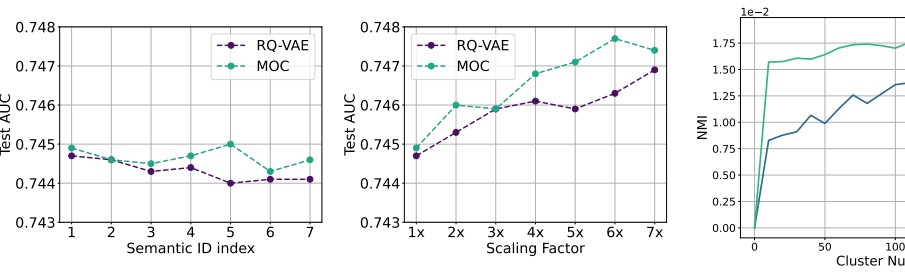

(a) AUC w.r.t. Single ID.  (b) AUC w.r.t. Multi ID.

Figure 8: More comparison results with RQ-VAE.

Figure 9: Discriminability scalability of MoC Fusion.

Table 2: Ablation on MoC Fusion. The experiments are conducted over Toys dataset with DeepFM as the backbone.

| Method | 2x | | 3x | | 7x | |
|---|---|---|---|---|---|---|
| | w/o | w/ | w/o | w/ | w/o | w/ |
| RQ-VAE | 0.7409 | 0.7414 | 0.7405 | 0.7407 | 0.7398 | 0.7413 |
| MoC | 0.7409 | 0.7408 | 0.7404 | 0.7415 | 0.7416 | 0.7418 |

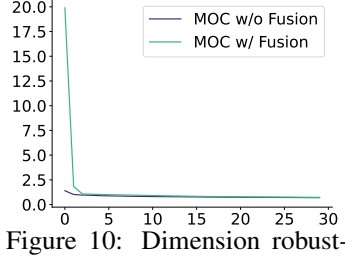

Figure 10: Dimension robustness scalability of MoC Fusion.

robustness scalability of MoC Fusion. In Fig. 9, it can be observed that the fusion module enhances the overall discriminability scalability of MoC by mixing the features. In Fig. 10, we truncate the singular spectrum and surprisingly find that the fusion module amplifies the principal components of the Semantic Representation, resulting in significantly higher top singular values. Additionally, the long-tail part is close to that of RQ-VAE and does not suffer from dimension collapse.

## 6 RELATED WORK

**Discrete representation learning.** Discrete representation is first introduced by (Van Den Oord et al., 2017) and shows feasibility and great success in the field of generate models (Esser et al., 2021; Lee et al., 2022; Rombach et al., 2022; Peebles & Xie, 2023; Mentzer et al., 2023). It utilize vector quantization (VQ) to model distributions over discrete variables with a codebook and define a simple uniform prior instead of Gaussian prior in VAE (Kingma, 2013) to avoid posterior collapse. RQ-VAE (Lee et al., 2022) further introduces residual quantization for better minimization of reconstruction error. FSQ (Mentzer et al., 2023) remove auxiliary losses and replace the vector quantizer in VQ-VAE with a simple scalar quantization.

**Semantic IDs.** The discrete representation obtained through VQ-VAE can be employed as a semantic clustering IDs, thus capturing the local structure of the LLM embedding to a certain extent. In the context of the recommendation system, TIGER (Rajput et al., 2024) takes advantage of a hierarchical quantizer (Lee et al., 2022) to convert items into tokens for generative recommendation and retrieval. LC-Rec (Zheng et al., 2024) improves TIGER by incorporating knowledge from LLMs like LLama (Touvron et al., 2023) and introducing instruction tuning tasks for effective adaptation to recommender systems. LMINDEXER (Jin et al., 2023) learns the Semantic IDs in a self-supervised styles to obtain the document's semantic representations and their hierarchical structures.

## 7 CONCLUSION

In this paper, we investigate the scalability of semantic representation based on LLM for recommendation. We unveil that simple methods, such as using a single codebook with multiple embeddings and scaling with hierarchical codebooks in RQ-VAE, do not scale effectively in Semantic Representation. We propose a novel multiple codebooks method which learn multiple independent Semantic IDs in the VQ-VAE based on LLM embeddings, and then employ these codebooks with a fusion module in the downstream recommendation models. Comprehensive experiments show that the proposed method successfully achieves scalability regarding discriminability, dimension robustness, and performance.

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
