# OpenReview forum: "Towards Scalable Semantic Representation for Recommendation"
_ICLR.cc/2025/Conference — ICLR 2025 Conference Withdrawn Submission_

### Official Review · Reviewer_uLBG · 2024-11-01

**Soundness:** 3
**Presentation:** 4
**Contribution:** 3
**Rating:** 6
**Confidence:** 4

**Summary:**

It is still a challenge to incorporate the great power of LLMs into enhancing recommender systems. Simply compressing the the high-dimensional nature of LLM embeddings to fit the smaller ID embeddings of recommendation will lead to information loss.
Following the multi-embedding setting and VQ-VAE, this paper proposes a two-stage Mixture-of-Codes generating semantic IDs by
multiple codebooks in a VQ-VAE framework, and then fuse them using a fusion module in the downstream recommendation stage. The performance table and visualization further verify its effectiveness and scalability.

**Strengths:**

1. Compared with the hierarchical design of RQ-VAE, MoC develops a parallel framework with multiple codebooks.
2. The study conducts intensive visualizations and experiments on three public datasets (Amazon Beauty, Sports, and Toys) and compares MoC with existing approaches.

**Weaknesses:**

1. The 3rd paragraph of Introduction section seems like the paraphrase of the 1st paragraph of section 3.1.

$\textbf{Introduction: }$"Notably, the LLM embeddings usually have very large dimensions, ranging from 4,096 to
16,384 (Dubey et al., 2024). When generating the embeddings for these codes, their dimension needs
to match that of the recommendation IDs. However, the dimension in recommendation is usually
small due to the Interaction Collapse Theory (Guo et al., 2023)."

$\textbf{Section 3.1: }$"The original LLM embeddings span a high-dimensional space, e.g., ranging from 1,024 to 416,384
dimensions. (Dubey et al., 2024). When we build a semantic representation based on the LLM embeddings to transfer their knowledge to recommendation, the dimension of these representation needs
to match that of the recommendation ID embeddings. However, the dimension of recommendations
is usually no more than 256 due to the Interaction Collapse Theory (Guo et al., 2023), and hence is
much smaller than that of the LLM embeddings. "

I think the illustration of section 3.1 will lead to redundancy and therefore not needed.

2. The illustration on fusion module is not clear. It is just a self-gating network. Did authors compare with other fusion experiments (e.g., concatenation, single MLP, or cross-attention)?

**Questions:**

1. what are x-axis and y-axis in Figure 10?
2. In section 4.1, average over quantized embedding is introduced into loss function for a shared VAE decoder. What about train different VAE decoder for each codebook?

---

### Official Review · Reviewer_Nrct · 2024-11-03

**Soundness:** 1
**Presentation:** 1
**Contribution:** 2
**Rating:** 3
**Confidence:** 5

**Summary:**

The paper first conducts experiments to demonstrate that ME (multi-embeddings) and RQ-VAE are not scalable approaches for semantic representation. The author(s) then introduce mixture-of-codes, which learns multiple parallel codebooks as semantic representations and employs a fusion module to capture feature intersections. Experiments are conducted on three public datasets to demonstrate the effectiveness of the proposed approach.

**Strengths:**

1. Timely study on semantic IDs, semantic representations, and scaling law in recommender systems.
2. Experiments are conducted on three public datasets.
3. The idea of calculating mutual information as an indicator of discriminability is interesting.

**Weaknesses:**

1. Taking RQ-VAE-based semantic IDs for semantic representations is impractical.
    1. RQ-VAE-based semantic IDs are structured to maintain sequential dependence across different levels, making them commonly used in generative retrieval or generative recommendation models. In these cases, an autoregressive model is trained to capture these semantic IDs in a sequence-to-sequence fashion. From my understanding, RQ-VAE is not well-suited for direct application as features or semantic representations for tasks like CTR. Additionally, I have not found any papers that describe the use of RQ-VAE-based semantic IDs as features (representation learning) for ranking, either in the references of this paper or in my own literature review. The author(s) cite several papers to suggest that RQ-VAE is widely used for deriving semantic IDs in recommender systems; however, the cited methods apply these semantic IDs in a fundamentally different way than in this paper (specifically, as features for ranking). Therefore, RQ-VAE-based semantic IDs appear to be unreliable as one of the two most important base methods for both analytical and performance experiments.
    2. When discussing RQ-VAE-based semantic IDs, I’d like to suggest two alternative approaches for learning semantic representations.
        1. Product/Vector Quantization [1]: This method can also encode input vectors into multiple semantic IDs but without sequential dependencies. It’s been used in existing literature as a semantic representation approach. Vector quantization is much simpler than the RQ-VAE-based method and is worth exploring further.
        2. Multiple Adaptors on LLM Embeddings [2, 3]: Instead of using semantic IDs, this approach directly maps input LLM embeddings to the recommendation space through adaptors (e.g., MLP). The scalability of this method can be explored by increasing the number of adaptors.
    3. In Section 3.1, the author(s) discuss the limitation of a single code. However, few methods actually use a single code. For example, TIGER uses 4 codes per item, and VQ-Rec [1] uses 32 codes per item. It would be better if more practical settings were adopted, like 4 -> 32 instead of 1 -> 3 semantic IDs, when drawing the observations in Figure 1.
2. Experiments are not solid enough.
    1. In Table 1, only methods using additional semantic representations (e.g., ME, RQ-VAE semantic IDs, MoC) are compared. Including a baseline method that relies solely on the original features, without any added semantic representations, would provide a clearer sense of the value these scalable semantic representations offer.
    2. Does "1x, 2x, 3x, 7x" in Table 1 mean "1, 2, 3, 7" levels of semantic IDs (or embeddings of ME)?
    3. It appears that there is no indication of repeated experiments or statistical significance testing, especially since the AUC metrics do not show large differences across all methods.
    4. Ablation study (Table 2).
        1. The effectiveness of applying the proposed fusion method on MoC seems not significant. On settings "2x" and "7x", the fusion module doesn't bring gains for MoC but downgrades the performance.
        2. Only one dataset is used for ablation study.
3. Presentation issues.
    1. In line 124, "416,384" -> "16,384"?
    2. In Eqn. (3) and (4), $h$ and $n$ are not defined. What are the connections between $n$ and $N$?
    3. In lines 201 - 202, what does "quantized representation $Q(r)$" actually mean? Does it mean the clustering centroids of RQ?
    4. What are the references of "a robust semantic representation should have higher top singular values" in Definition 3.2?
    5. Eqn. (8) doesn't match Figure 6. There is no activation function between $W_{down}$ and $W_{up}$ in Eqn. (8).
    6. Line 244, "low-frequency information" and "high-frequency" are not defined in the paper.
4. Code is not available during the reviewing phase.

[1] Learning Vector-Quantized Item Representation for Transferable Sequential Recommenders. WWW 2023.

[2] Where to Go Next for Recommender Systems? ID- vs. Modality-based Recommender Models Revisited. SIGIR 2024.

[3] Towards Universal Sequence Representation Learning for Recommender Systems. KDD 2022.

**Questions:**

Please refer to "Weaknesses" for details.

---

### Official Review · Reviewer_4wGE · 2024-11-03

**Soundness:** 2
**Presentation:** 2
**Contribution:** 2
**Rating:** 5
**Confidence:** 3

**Summary:**

The authors find when applying LLM embeddings to recommendation systems: the high dimensionality of LLM embeddings must be reduced to fit the lower-dimensional space used in recommendation models, leading to information loss. To solve this problem, authors propose a two-stage method called "Mixture-of-Codes" (MoC), which uses multiple indepentent codebooks for vector quantization of the LLM embeddings, then utilizes the Semantic representation for downstream task. This approach aims to improve both the discriminability and dimension robustness of semantic representations, ensuring that more detailed semantic information is preserved.

**Strengths:**

1. The paper is well-written and easy to understand.
2. The paper pioneers as study on the scalability of semantic representation on transferring knowledge from LLM to recommendation systems.

**Weaknesses:**

1.	The current experimental design does not sufficiently validate the authors' claims

   a)	The improvements are not significant. In table1, compared to the non-scaling version, the improvement is often only around 0.1%.

   b)	The necessary baseline comparison using IDs to scale embeddings without using codebooks is missing. If there is no significant improvement compared to this baseline, then there is no need to explore the use of semantic IDs.

   c)	Contradic to this paper's claim: MoC will help for scaling up, using the MoC method leads to a decrease in AUC metrics on some datasets. (i.e. in Table 1, model DCNv2, Toys dataset, and model AutoInt+, Beauty dataset). Moreover, MoC does not consistently outperform the baseline, and in some cases, the RQ-VAE method achieves the best metrics. (i.e. in Table 1, model DeepFM, Beauty dataset, 7x).

   d)	The components in MoC do not always contribute positively. removing the MoC fusion module actually led to metric improvements in certain cases. (Table2, MoC method, 2x)

   e)	Lack of ablation of main component of MoC. There lack a fair comparison with ME [1] with fusion.

2.	The mathematical definitions in the article could be further improved.

   a)	There is a lack of definition for 1_{x_1} indicates the one-hot encoding of x_i in X_i in lines 096 and so does line 150

   b)	In lines 321, The authors have omitted the mathematical modeling of the activation layer.


[1] Guo, Xingzhuo, et al. "On the Embedding Collapse when Scaling up Recommendation Models." Forty-first International Conference on Machine Learning.

**Questions:**

See Weaknesses.

If the authors address these concerns, I can consider a higher rating.

---

### Official Review · Reviewer_8hgy · 2024-11-05

**Soundness:** 3
**Presentation:** 2
**Contribution:** 3
**Rating:** 6
**Confidence:** 3

**Summary:**

This paper investigates the scalability of semantic representation based on LLM for recommendation. The authors introduce an innovative approach termed Mixture-of-Codes, which involves the creation of multiple independent codebooks during the indexing phase. Subsequently, it employs semantic representations in conjunction with a fusion module during the recommendation phase. Comprehensive experiments demonstrate that this proposed method effectively achieves scalability in terms of discriminability, dimensional robustness, and overall performance.

**Strengths:**

1.	This paper discusses an interesting problem: how to scale up semantic representations on transferring knowledge from LLM to recsys, and provides a detailed analysis of the shortcomings of some traditional approaches.
2.	The authors propose an effective algorithm (Mixture-of-Codes) to improve the scalability of discrete semantic codebooks, and the experimental results validate the effectiveness of the proposed method.

**Weaknesses:**

1.	The writing of this paper needs to be improved. For example, Sections 4.1 and 4.2 use the same symbol N to represent two different variables, which seems inconsistent. Besides, there are many experiments and analyses through the paper, and the authors should pay more attention to the relationships between them.
2.	The use of discrete semantic codebooks in this paper is innovative, but before the authors further discuss how to use them better, they should first elaborate on the advantages and benefits of the basic method in detail (please refer to question 2).
3.	This paper aims to scale up semantic representation, but the authors do not explain why scaling up is necessary and what’s the significance. I agree that “scaling up” could be impactful, but the authors should provide a detailed discussion.

**Questions:**

1.	To my best knowledge, this paper discretizes item text representations as additional discrete features to assist the CTR model learning, rather than directly transforming and incorporating the raw features extracted from LLM (e.g., through adapters) into the model. If I have understood this correctly, why do the authors emphasize various issues in the third paragraph of the introduction regarding the transition from high-dimensional LLM features to low-dimensional recommendation features? Since the core idea is "feature engineering”, even if the dimensionality of the features from LLM and the recommendation model is the same, these issues should still exist.

2.	(Followed by Question 2 and Weakness 1) The author's approach of using quantized semantic codebooks as additional features is innovative; it differs significantly from most works that use discrete codewords as item identifiers in generative recommendations. Therefore, I believe that the author should emphasize the advantages of this approach compared to not adding these features. For example, I think it is necessary to include the comparison of the proposed method with the original method that does not utilize discrete representations (but it does not provide results for this basic baseline). Additionally, the benefits of adding these features should be further analyzed.

---

### Author Response · Authors · 2024-11-25

We would like to express our gratitude to all reviewers for taking the time to engage with our work. However, after careful consideration of the feedback and scores received, we have decided to withdraw our paper.

---

### Note · Authors · 2024-11-25

I have read and agree with the venue's withdrawal policy on behalf of myself and my co-authors.